# Reproducibility Study on Curriculum by Smoothing

## Reproducibility Summary

"Curriculum by Smoothing" introduces us to the idea of adding curriculum to a convolutional neural network, by smoothing out the results from each convolution layer. The author inspects the performance of their proposed method by comparing baselines with and without CBS. They ran several experiments on different variants of the CNN architectures. They targeted the image classification task, feature extraction, and the transfer of the learned model to a different task. They further evaluated their models on zero-shot domain adaptation and generative models to test their solution's generality.

In our reproducibility study, we will run the same set of experiments and reproduce the observations. Furthermore, we will contrast our findings with the ones documented in the paper. As outlined in the quantitative results from the paper, the comparison reveals improvements over the CNN models trained without using CBS. We plan to verify this observation and also provide any valuable insights discussing the reproducibility aspect of the proposed technique.

**Scope of Reproducibility**

The author's central claim is that, adding a Gaussian kernel to the output of each convolution layer smooths the feature map and ultimately reduces the noise and distorted artifacts; which eventually help during the early phase of training a deep neural network in better generalization. So in our reproducibility plan, we are mainly focused on the core idea of curriculum learning by adding smoothness using the Gaussian kernel.

**Methodology**

The author's code was not available, so we implemented the original architecture first, referring to the papers for setting the model parameters and regularization. Then we added the Gaussian layer after every convolution layer to match the implementation from the paper.

**Results**

We ran our experiments on the image classification task and observed that the central claim by the author hold true in our case as well.

**What was easy**

Standard baseline CNN variants were used for experiments, therefore, it was easy to implement most of the network.

**What was difficult**

The most challenging part of the model was the implementation of Gaussian kernel, as we had to dive deeper into the Backend for extracting the right operation to implement the method. Also, we have to change the value of $\sigma$ after some epochs, for that we had to design a custom Gaussian layer function.

**Communication with original authors**

We had some issues implementing the method as described in the paper. For this we communicated with the authors through mail. Communications with the author are detailed at the end of this report.

# 1 Introduction

Curriculum learning is the dataset preprocessing technique used to train deep neural networks where the core idea is to sort datasets according to their difficulty level. During the initial phase of training, we expose the deep network to easy examples and gradually increase the difficulty of examples. This idea's primary motive is the faster convergence of deep networks and better generalization of the task. During the network's early training phase, information propagated can contain distorted artifacts due to noise that can disturb the training.

In the paper, the author proposed a curriculum-based technique that smooths feature embedding of a CNN using anti-aliasing or low-pass filters. Their main idea is controlling the high-frequency information propagated during the training of CNNs by convolving the output of a CNN feature map of each layer with a Gaussian kernel. As the Gaussian kernel variance decreases, it increases the amount of high-frequency information available within the network for inference. As the amount of information in the feature map increases, the network can learn to represent the data better progressively.

# 2 Scope of reproducibility

The author extends the applications of adding CBS in the following three scenarios:

1.) Better task performance: How does the model's accuracy vary when trying with curriculum and without a curriculum learning approach?

2.) Better feature extraction: If we use the trained model to extract the features from different datasets to train a weak classifier and for various vision tasks where pretraining is required, how does the model's performance vary?

3.) Generative Models: How does the curriculum-based smoothing help in different vision tasks which utilize CNNs such as generative models?

We are mainly focusing on the author's first claim and try to show a comparison of without curriculum and with curriculum difference in the accuracy of the results of the various architecture of the deep neural network. However, in a future update, we also want to perform some more experiments mentioned in the future work section below.

# 3 Methodology

The proposed method in [1] talks about adding a curriculum to the CNN and its variants(or the models that employ convolutions to extract features from data and perform some downstream tasks). A Gaussian kernel is added after every convolution to smooth out the features and allow low-frequency information to pass on. Since the code for [1] was not available, we implemented it from scratch. We developed our program in Tensorflow, as it was mentioned in the paper that the native code for [1] is in Pytorch. Overall we conducted 36 experiments and report all the results in Table 1. In the subsequent sections, we talk about our implementation for the CBS when added to the baseline models.

## 3.1 Model descriptions

In the paper, the author used the variants of some popular CNN architectures such as VGG-16, ResNet-18, Wide-ResNet-50, and ResNeXt-50. These models are also used as baseline and the CBS variants for each of the models is developed by adding a layer that performs Gaussian smoothing after every convolution.

We focused most of our study on figuring the implementation details, and the addition of Gaussian filter. Therefore, we limited our tests to VGG-16 and ResNet-18, particularly to the image classification task. In these models, like discussed in [1], we add the Gaussian filtering layer is added after each convolution layer.

The authors used the same model configuration and hyper parameters as delineated in the original paper of the model. We also retained some of these configurations such as weight decay and batch size, but we decided to tweak some of the hyperparameters to get decent validation accuracy. The analysis on these hyperparameters are provided in the upcoming section.

The first set of experiments is of classifying images using SVHN, CIFAR -10, and CIFAR - 100 dataset with and without CBS using the models mentioned above.

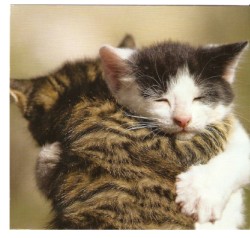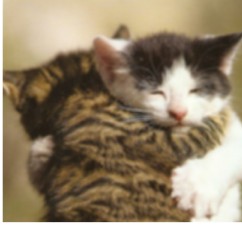

Figure 1: The left image is the original one, which we passed to our single layer, Gaussian filtering model. The layer was added just after the input. By obtaining a blurred image(right) we ascertained the functioning of our Gaussian layer.

## 3.2 Datasets

The datasets used in the image classification task are SVHN, CIFAR-10, CIFAR-100, and ImageNet. For the second experiment of Feature Extraction, they have used SVHN, CIFAR-10, and CIFAR-100 dataset. For the third experiment of Transferring to Different Task, they have used MNIST and USPS datasets.

In our experiments we use the SVHN, CIFAR-10 and CIFAR-100 datasets repectively to image classification task. All three of these datasets were available in the Tensorflow datasets module, so it made simple and efficient to organize and load the [train, tests] sets.

Below are the source of each of the dataset that we used,

1.) SVHN from `https://www.tensorflow.org/datasets/catalog/svhn_cropped`
2.) CIFAR-10 from `https://www.tensorflow.org/datasets/catalog/cifar10`
3.) CIFAR-100 from `https://www.tensorflow.org/datasets/catalog/cifar100`

## 3.3 Hyperparameters

We can observe from the table 1 that our result for the experiments are lower in comparison to the results reported by the authors. During the training phase, we required our baseline model to perform better and for the time being we were not able to implement the learning rate scheduler to monitor on validation accuracy(decay on plateau); so we tried and tested different values. Then we trained the CBS counterparts on the same hyperparameters used in the original ones. We used the Adam optimizer with its learning rate set to 0.001 or 0.0001 initially. In the original implementation, they used a learning rate($lr$) decay by the factor of 10 when test accuracy is stagnant or decreased $lr$ by the same factor on a set number of iterations.

Additionally, we also examined hyperparameters concerning the CBS model, such as decay rate, $\sigma$ and the kernel size for the Gaussian layer function. We communicated with the author and he quickly resolved our queries regarding kernel size. We set the kernel size to $(3, 3)$ and $\sigma$ to 1 or 2 with a decay rate of 0.9 after the completion of 5 epochs, as mentioned in the paper [1]. All the experiments that we conducted were carried out for number of epochs within 100.

## 3.4 Implementation details

We carried out the implementation for the model in three phases:

**Phase I:** First we developed the baseline model, and also arranged the dataset. We used the Tensorflow dataset library(linked before in the datasets section) for loading ready to use datasets in our module.

**Phase II:** Next and the most crucial part was to develop the Gaussian filter layer for smoothing out the features from each convolutions.

For our initial implementation with Keras, we used callbacks to call and change the standard deviation in the Gaussian Noise layer. But, we received bad results; in fact, later when we delved more into this, we discovered that there Keras have two layers that already have been implemented using Gaussian noise – Gaussian Noise(additive) and Gaussian Dropout(multiplicative). We had accidentally incorporated the Gaussian Noise layer, which did the opposite of smoothing(i.e. it adds a noise sampled from a standard normal distribution). In the end, we designed our own custom function to design a kernel and performed depth wise convolutions with the source.

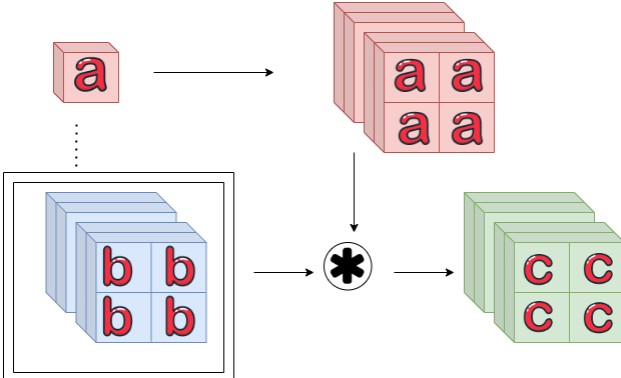

Figure 2: The pipeline for the Gaussian layer function, where we take a kernel of specified shape(a in figure). Assuming we have an input matrix(b), a new kernel is returned with shape of the kernel "a" expanded and conformed to the shape of "b". Next both these matrices are convolved depth wise(2d convolutions on the first channel, then second channel and so on; Also, the double boundary around "b" indiacates 'same' padding). Finally, we get a matrix with smoothed out features.

**Phase III:** Finally, before moving to the experimentation, we also carried out sanity check to see if our custom Gaussian layer function smooths out the input or not. For this we applied our layer function on an image, as shown in the Figure 1.

Next, we move to the experimental setup where we explain our procedure briefly. However, let us first take a look at our implementation of the Gaussian layer.

**The Gaussian Layer function** The Gaussian layer function is a custom layer designed to perform depth wise convolutions on an input matrix. According to our understanding of the paper, the layer acts like Gaussian low pass filter with the following functions:

$$G_\sigma = \frac{1}{2\pi\sigma^2} exp \left( -\frac{x^2 + y^2}{2\sigma^2} \right)$$

We implement this function as a utility function that returns a kernel of size $(3, 3)$ to call it inside another function that forms a depth kernel by expanding and repeating along dimension(the channel). Then we call a depth wise convolution operation from Tensorflow Backend to compute the convolved value for each channel. This is illustrated in the Figure 2.

In this filter we also pass the sigma value as a variable, which will decay with a specified rate (we too assumed the decay rate 0.9, as in the paper).

## 3.5   Experimental setup

Our implementation was broadly divided into two tasks, one for the baseline model and the other for the proposed model(baseline with CBS) from the paper. We latched one process on one of the GPUs (Geforce RTX 2080) and the other on the other GPU (Geforce RTX 2070). At run-time the program downloads the required datasets and stores them in the "datasets" folder for future use. We also set up a checkpoint directory for saving models with best validation accuracy for each experiment. Each individual experiments were conducted for less than100 epochs for 3 different seeds(5, 10 and 15). We report the mean validation accuracy along with standard deviation across different seed for experiments on datasets on corresponding models. We report our results in table 1.

Our evaluation is lower in validation score in some datasets and models. The reason for such a low score is firstly, we used an Adam optimizer with no control over the learning rate. We conducted the same experiment with SGD as well (exponential decay on lr), but Adam was performing better and also the original model implemented lr decay either on plateau(stagnant validation accuracy) or in a fixed number of iterations. But, for implementing the Gaussian kernel layer as a function we had to open up the architecture, carefully lining up the function with the convolution layers to simulate CBS.

Alternatively, we also implemented the models and called them via Keras "fit" method(our first preference, and also achieved better accuracy in this way) but by doing this, we had the limitation of only accessing the models functions/layers via "callbacks" (which only complicated the implementation more).

Table 1: Results of our experiments on Image classification via CNN variants with and without CBS. We report the mean and standard deviations of the validation accuracy for each of the models with its corresponding datasets over 3 different seeds(5, 15, 20).

|  | SVHN | CIFAR10 | CIFAR100 |
|---|---|---|---|
| VGG16 | $94.7 \pm 0.15$ | $72.68 \pm 0.49$ | $34.95 \pm 0.16$ |
| VGG16(CBS) | $93.46 \pm 0.28$ | $75.59 \pm 2.36$ | $39.24 \pm 0.28$ |
| ResNet18 | $90.64 \pm 0.27$ | $64.5 \pm 0.35$ | $26.5 \pm 1.7$ |
| ResNet18(CBS) | $93.53 \pm 0.39$ | $72.51 \pm 0.28$ | $36.74 \pm 1.13$ |

## 3.6 Computational requirements

The different sets of experiments that we conducted took different times Each of the experiments took roughly 60 minutes to run on GPU. The CNN models with CBS took longer as we supplement an additional layer of Gaussian kernel after each convolution. The ImageNet dataset was also used in the author's experiments; however, we did not carry out any experiment on the ImageNet dataset given the present computational resources.

The time required by the baseline was less than CBS one, since we had more computation to carry out in CBS. The way we implemented we had to calculate Gaussian kernel at each call of the forward( for future plans, we don;t know how authors did but this part could be improved further)

## 4 Results

Our results support the claim presented by the authors in [1], i.e. curriculum via Gaussian kernels at the end of each convolution layer increases the models performance. We observe that there an increase in validation accuracy for all those baseline models where CBS was added. We kept our epochs less than 100 and also clipped those which took longer to train (like SVHN and CIFAR100). We did this in order to verify the central claim with a slight constraint on experiments(on number of epochs and number of seed).

**Results on Image classification**

Table 1 shows the results from our experiments on models in the Image classification task. For VGG16 we set the initial value of $\sigma$ to 2 and decreased it with a decay rate of 0.9 every 5 epochs. We used Kaiming initialization(as in the paper) for initializing the weights of each of the layers, and also added weight decay via l2 regularization. For the ResNet18 case we initialized $\sigma$ with 1 and decreased it with the same factor.

Also, in our experiments we see that the VGG16 model when trained on SVHN dataset performed equally(slightly higher) with respect to the VGG16 model with CBS added.

Overall, the models with Gaussian smoothing added to each convolution layer perform better than the baseline.

## 5 Discussion

Throughout the study, we gained more and more insights on the functioning of the method proposed in the paper. Also the challenging aspects of the paper, which in most deep learning literature is the knowledge of how a learning paradigm works in practice. The authors did a great job in getting through their idea in an intuitive way.

### 5.1 What was easy

The proposed model is a simple addition of a Gaussian kernel after each convolution layer. All the concepts communicated in the paper are fundamentals that we knew about and was easy to relate to. Also the baseline models are the standard CNN variants used, so, it was easier to implement from scratch.

### 5.2 What was difficult

Like we discussed in the last subsection, implementing the baseline models was easy, but, training them efficiently and the hyperparameter tuning made it a bit difficult for us to reproduce results at full potential(which we believe could have been done by increasing number of epochs, playing with the batch size and tuning the learning rate).

Moreover, some of the operations required core knowledge of the tensor and Backend operation. And also to understand through visualization and analysis if the operations(in the Backend) that we are using, produce the right results or not.

## 5.3 Communication with original authors

We contacted the corresponding author in the paper and asked a few doubts through email.

**Doubt 1:** How adding kernel layer after convolution layer is different here? As anyway, there is a pooling layer doing downsampling after the convolution layer.

**Authors' clarification:** "We add the Gaussian filter after each convolutional layer. Say you have 10 CNN layers in a model (you won't have ten pooling/downsampling layers, of course), so we add the gaussian kernel after each of those layers."

**Doubt 2:** This is a random thought. In the case of neural network architecture, if we add noise to the layer's weight, it acts as a regularizer. It behaves as an L2 penalty to try to control overfitting and provide better generalization. In this paper, we are adding noise in the convolution layer's output, so can we say this acts as a data regularizer of regularizing the convolution layer output for reducing overfitting. Therefore, it generalizes well and improves accuracy, or is it curriculum helps in generalization here? Do we get faster convergence here in the case of the curriculum?

**Authors' clarification:** "I am not sure if this would reduce the overfitting problem; at least, I don't have much empirical evidence suggesting that that would be the case. What is happening is that you are progressively adding information that is propagated within the network. One way you can loosely relate it to overfitting is that you train models that don't try to overfit/fit spurious features/noise in the data since that noise will be smoothed out. As the network learns more, you can assume that the amount of noise in the features will decrease, so we can "safely" add more information within the network (reducing the variance of the kernel). There was no real discernable difference in the "best epoch accuracy"."

**Doubt 3:** Is this method of the curriculum by smoothing can apply other than image data? If yes, how can we make it applicable in different types of data such as text or RNN based models?

**Authors' clarification:** "I strongly think that it can apply to more than just image data! That's something I am quite interested in working on. Especially in the case of RL, perhaps? "

**Doubt 4:** Does this method depend on the dataset, or is it, dataset independent?

**Authors' clarification:** "We worked with a variety of datasets and types of tasks (image classification, VAEs, 0-shot domain adaptation, transfer learning). These make me think it's not super dependent on the dataset."

**Doubt 5:** Did the Wide-Resnet 50 used in experiments, had a widening factor of 2.

**Authors' clarification:** "I tried to find this information, but I can't seem to might need a bit of digging."

**Doubt 6:** For ResNext 50, what was the bottleneck width?

**Authors' clarification:** "I believe the cardinality was 8."

**Doubt 7:** What is the size of the Gaussian kernel?

**Authors' clarification:** He used Gaussian kernel of size 3 for his experiments.

# 6 Future Work

In the future, it is interesting to check the anti-curriculum concept on a different task to see how it behaves. It is also interesting to apply the Laplacian of Gaussian filter as a curriculum instead of only Gaussian kernel to see how it behaves. We plan for convergence timing experiments with the curriculum by smoothing and without curriculum by smoothing on various tasks. Also, currently we programmed the network in a way that it is convenient for us to add the Gaussian smoothing function. This way our code for ResNet18 contained lot of lines, which made it harder to follow and expand to more layers. We plan to work on a more efficient design on these models so that people will have access to the program in Tensorflow.

## Code Repository of our experiments

https://github.com/jaideepbankoti/reproducibility2020-cbs

## Acknowledgement

We thank Samarth Sinha (Corresponding author) for helping us with various details and clarification of doubts, and we acknowledge the kind of support he gave us to reproduce the experiments.

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
