# OpenReview forum: "Reproducibility Study on Curriculum by Smoothing"
_ML_Reproducibility_Challenge/2020 — Reject_

### Official Review · AnonReviewer1 · 2021-02-09
**The authors released the code**

**Rating:** 6
**Confidence:** 3

**Review:**

This reproducibility report is relatively easy to follow. Their re-implementation supports the claims in the original paper.

At the time of writing this review (2021, Feb 8), the authors of the original paper released the code. This could make reproducibility easier.

**Familiar With The Original Paper:**

I have not read the original paper

**Reproducibility Summary:**

Report has summary

---

### Official Review · AnonReviewer2 · 2021-03-02
**RE: Reproducibility Study on Curriculum by Smoothing**

**Rating:** 5
**Confidence:** 4

**Review:**

In this paper, the authors discussed their attempt to reproduce the results reported in [1].

Pros:
1), implemented the method proposed in the original paper from scratch using a different framework, i.e., Tensorflow. Pytorch was used in the original paper.
2), successfully demonstrate the value of adding the Gaussian filtering layer during curriculum learning comparing to the baseline model. One thing that the authors should make it clear is that whether the baseline model was also trained with curriculum.

Cons:
1), only performed reproducibility study of a small part of the results reported in the original paper.
2), the reported performance even though demonstrates the benefit of CBS, but does not reach the level reported in the original paper.
3), very limited study of hyperparameter search
4), no ablation study


**Familiar With The Original Paper:**

I have read the original paper

**Reproducibility Summary:**

Report has summary

---

### Decision · Program_Chairs · 2021-03-31

**Decision:**

Reject

**Comment:**

Overall reviews and/or the paper content not good enough for the AC to recommend to the journal.